# LoD: Unlocking Performance Gains in Compression via Differential Analysis

## Abstract

Mixed-precision compression reduces model size and enhances inference efficiency, yet mainstream research focuses on minimizing accuracy loss from parameter compression. However, experimental evidence and observations often reveals performance improvements under specific conditions, challenging the assumed performance-efficiency tradeoff. These gains, often attributed to fortuitous alignments, lack systematic explanation and exhibit instability in performance across models and datasets. This work investigates these phenomena using a loss-driven framework based on total differential analysis, addressing three interconnected questions: **(1)** What conditions enable mixed-precision compression to enhance performance? **(2)** How can we model and control performance instability to ensure lossless outcomes? **(3)** What are the theoretical boundaries for achieving lossless compression? We take two mainstream compression methods as examples, parameter decomposition and quantization and propose a loss-driven(LoD) theoretical framework. For decomposition, we optimize layer-wise ranks within lossless neighborhoods. For quantization, we formulate compression as a grouped knapsack optimization problem. Extensive experiments across diverse datasets and architectures validate consistent, stable gains. And the code will be released.

## 1 Introduction

With the rapid growth in scale and complexity of deep neural networks, demands on memory and computational resources have surged, making model compression a critical technique for accelerating inference and reducing deployment costs. Among various approaches, post-training compression has gained popularity due to its low overhead and compatibility with existing training pipelines. Within this context, mixed precision compression stands out: it applies heterogeneous compression strategies across layers—such as assigning different quantization bit-widths Banner et al. (2018); Liu et al. (2021a); Hu et al. (2023); Zhang et al. (2024) or decomposition Bisgard (2020) ranks according to layer sensitivity—to flexibly allocate resources while preserving accuracy. Representative approaches include multi-point quantization that linearly combines discrete values to approximate full-precision weights Liu et al. (2021a). AWQ Lin et al. (2024) which jointly calibrates weights and activations to improve low-bit robustness. Together, these methods demonstrate the potential of mixed compression to achieve an optimal performance–efficiency trade-off.

Conventional wisdom views compression as inherently lossy, implying an unavoidable trade-off between efficiency and accuracy. However, emerging empirical evidence challenges this paradigm, revealing that compression can sometimes preserve or even enhance accuracy. For instance, as shown in Fig. 1, DAC Li et al. (2019) reproduces baseline accuracy after decomposing convolutional layers; MAESTRO Horváth et al. (2024) achieves a 0.72% gain on ResNet-50 via low-rank ordered decomposition; and RQ Louizos et al. (2019) and CET Zhang et al. (2025) employs bit-allocation strategies to surpass baselines in some cases. These methods highlight compression's potential to refine parameters under similar compression rates, improving generalization in specific settings.

Yet, this serendipity masks a deeper issue: the underlying mechanisms remain largely unexplained, with academia often attributing such gains to fortuitous alignments or unexplained anomalies rather than principled processes. Moreover, these improvements are not universally stable, frequently varying across models, datasets or other conditions, which hinders reliable application. This instability raises a pivotal question: how can we systematically model and control it to transform empirical

| Method | Lossless Rate | Swin_T | BERT | ResNet₅₀ | ConvNeXt | VGG16 | ResNet₁₈ |
|---|---|---|---|---|---|---|---|
| DAC 2018 | 0/6 | ✗ | ✗ | ✗ | ✗ | ✗ | ✗ |
| RQ 2019 | 1/6 | ✗ | ✗ | ✗ | ✗ | ✗ | ✓ |
| Chenna 2023 | 1/6 | ✗ | ✗ | ✗ | ✗ | ✗ | ✗ |
| Zhang 2023 | 1/6 | ✗ | ✗ | ✗ | ✗ | ✓ | ✗ |
| Maestro 2024 | 1/6 | ✗ | ✗ | ✓ | ✗ | ✗ | ✗ |
| CET 2025 | 2/6 | ✗ | ✓ | ✗ | ✗ | ✗ | ✓ |
| LoD (Ours) | 6/6 | ✓ | ✓ | ✓ | ✓ | ✓ | ✓ |

Figure 1: The traditional trade-off between compression and accuracy has been shaken by recent findings. The chart on the right shows that existing methods can occasionally achieve "lossless" compression under certain conditions, but its stability and interpretability remain challenges.

luck into predictable outcomes? To address this, we propose LoD, a unified loss-driven differentiable framework, focusing on three key questions:

> 1) Under what conditions can mixed precision compression yield performance gains?

> 2) How can we effectively model the instability in compression performance? Is first-order analysis sufficient, or does second-order analysis better control these fluctuations?

> 3) How can we theoretically characterize the boundaries of lossless compression? At what compression levels can performance gains be achieved?

To answer these questions, we propose LoD, a model-agnostic, loss-driven differentiable framework. We then instantiate LoD on two representative mixed precision techniques, Tensor Decomposition: LoD integrates loss-preserving neighborhoods with low-rank constraints to automatically determine optimal per-layer ranks; Quantization: LoD reformulates the bit-width search as a grouped knapsack optimization within a lossless region. Empirical results demonstrate that LoD consistently enables performance improvements under both compression schemes, providing the first rigorous explanation for these elusive gains. Our contributions are as follows:

- We provide the systematic theoretical exploration of performance gains in mixed precision compression, shifting from empirical luck to predictable mechanisms.
- From differential neighborhoods, we formally delineate the scope and relative impact of first- and higher-order terms, offering analyzable boundaries for lossless compression.
- We propose a model-agnostic, loss-driven analytical framework, LoD, and apply it to parameter decomposition and model quantization. Empirical studies across diverse tasks and model architectures demonstrate its consistent effectiveness and broad applicability.

## 2 LOSS-DRIVEN ANALYTICAL FRAMEWORK

This section explores the mechanisms underlying performance gains from compression by addressing 3 questions.

### 2.1 UNDER WHAT CONDITIONS CAN MIXED PRECISION COMPRESSION LEAD TO PERFORMANCE GAINS?

Consider an $n$-layer neural network with parameters $w = (w_n, \ldots, w_1)$ and empirical loss:

$$f(w) = \frac{1}{m} \sum_{(x_i, y_i) \in \mathbb{D}} \ell(model_n(x_i, w), y_i), \quad model(x) = h_1(h_2(\ldots h_n(h_{n+1}, w_n) \ldots, w_2), w_1) \quad (1)$$

where $\mathbb{D}$ denotes the dataset, $m$ its size, $\ell$ the per-sample loss, and $h_i$ the network layers. This formulation is general and independent of the specific network architecture or compression method.

Compression techniques, such as quantization and decomposition, introduce perturbations to weights and activations. Accordingly, the post-compression loss for a sample is expressed as:

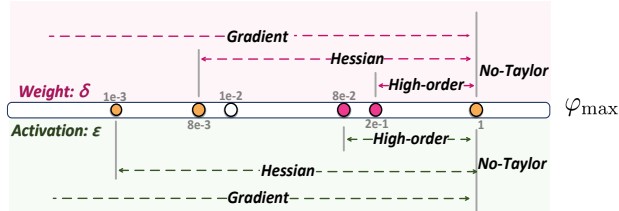

Figure 2: The figure shows three regimes in perturbation magnitude $\varphi$: Gradient (first-order), Hessian (second-order), and High-order / No-Taylor (higher-order). Markers are per-layer empirical thresholds $\varphi_{\max}$; beyond this scale higher-order terms cause performance instability.

$$\bar{\ell}(w, x_i, y_i) = L(h_1(h_2(\ldots h_{n-1}(h_n(h_{n+1}+\epsilon_n, w_n+\delta_n)+\epsilon_{n-1}, w_{n-1}+\delta_{n-1})+\cdots+\epsilon_1, w_1+\delta_1), y_i), \quad (2)$$

where $\delta_i$ and $\epsilon_i$ denote the compression-induced errors in $h_n$'s weights and its activations. This perspective enables the analysis of compression effects using perturbation theory and differential.

For any network layer $i$, assuming its activation and gradient vectors have bounded second-order moments, by applying the total differential, we obtain

$$\min_{\epsilon \in E} \bar{f}(w) - f(w) = \frac{1}{m} \sum_{(x_j,y_j) \in \mathbb{D}} \sum_{i=1}^{n} \frac{\partial \ell}{\partial h_{i+1}} \cdot \epsilon_i + \frac{\partial \ell}{\partial w_i} \cdot \delta_i + \frac{1}{2}(\epsilon_i, \delta_i)\mathbb{H}(\epsilon_i, \delta_i)^\top + O(||(\epsilon_i, \delta_i)||^3) \quad (3)$$

where $\mathbb{H}$ represents the Hessian matrix and $O(||(\epsilon_i, \delta_i)||^3)$ represents the high-order term, $\cdot$ is inner product and $*$ is the scalar product, $\bar{f}(w) = \frac{1}{m} \sum \bar{\ell}(\cdot)$. Eq. 3 directly links compression and model performance. Thus, we optimize the above expression to make $\bar{f}(w) - f(w) < 0$ to obtain the gain.

**Basis 1** *The total differential relies on a linear approximation assumption, valid only when the changes in the function's variables are sufficiently small.*

The constraint lies in the magnitude of compression-induced noise. To theoretically quantify the loss shift caused by perturbations, we define a local perturbation neighborhood $\mathcal{N}(x)$ that measures the discrepancy between the actual loss change and its total differential approximation.

**Definition 1 (Perturbation Neighborhood)** *For a given compression level $k$, we define the perturbation neighborhood as the discrepancy between the loss change and its differential approximation:*

$$\mathcal{N}(\theta_i^k) = \left| \tilde{\ell}(w + \theta_i^k, x_i, y_i) - \left( \ell(w, x_i, y_i) + \nabla\ell(w)^\top \theta_i^k + \tfrac{1}{2}(\theta_i)^\top \mathbb{H}\theta_i + O(||\theta_i||^n)) \right) \right|, \quad (4)$$

*where $\theta_i^k = \delta_i^k$, $\tilde{\ell} = \ell$ for weight perturbations, or $\theta_i^k = \epsilon_i^k$, $\tilde{\ell} = \hat{\ell}$ for activation perturbations.*

The parameter $k$ controls the compression level, such as bit-width (e.g., 4/8-bit) or rank. When $k$ denotes rank, larger values correspond to lower compression. The expansion in Definition 1 decomposes the loss shift into first-order (gradient), second-order (Hessian), and higher-order contributions, each scaling with the perturbation magnitude. The perturbation neighborhood $\mathcal{N}(\theta_i^k)$ thus quantifies how well the total differential approximates the true loss change under compression.

## 2.2 How We Effectively Model the Instability in Compression Performance?

Performance instability stems from nonlinear effects in Eq. 3, particularly the second-order term $,(\epsilon_i, \delta_i)^\top \mathbb{H}(\epsilon_i, \delta_i)$ and higher-order, which amplify loss fluctuations when perturbations exceed linear regimes. To answer the significance of higher-order terms, we first carry out a theoretical derivation. Specifically, we analyze perturbations of the separable form $\Delta = \varphi u$ (direction $u$, scalar magnitude $\varphi$). By differential expansion (Eq. 3) $\Delta L \approx \varphi\, g^\top u + \frac{1}{2}\varphi^2 u^\top \mathbb{H}u + R_3(\varphi, u)$, with $g = \nabla_w \ell$ and remainder $R_3 = O(\varphi^3)$. Requiring $\Delta L < 0$ yields the quadratic condition

$$\tfrac{1}{2}(u^\top H u)\varphi^2 - |g^\top u|\varphi + R_3 < 0. \quad (5)$$

Denote $a = |g^\top u|$ and $b = \frac{1}{2}u^\top \mathbb{H}u$. If $R_3$ is negligible the positive root gives the upper bound

$$\varphi_{\max} \approx \frac{2|g^\top u|}{u^\top \mathbb{H}u}, \tag{6}$$

Eq. 6 defines the maximal perturbation scale $\varphi_{\max}$ under which the first-order approximation dominates. When $\varphi > \varphi_{\max}$, the second-order curvature term becomes dominant and amplifies the performance fluctuations. To make this bound operational we estimate its ingredients on a small calibration set: $g$ is obtained by averaging per-layer gradients, $u^\top \mathbb{H}u$ is estimated via Hessian-vector products (or an empirical-Fisher proxy when $\mathbb{H}v$ is too costly). Plugging these estimates yields a numeric $\varphi_{\max}$ for each layer. Please see the Appendix for the complete algebraic derivation.

We then validate the analytic threshold by controlled perturbation experiments: for increasing perturbation magnitudes $\varphi$ (in the chosen norm) we record the proportion of the observed loss change explained by the first-order term. The results, summarized in Fig. 2, show distinct regimes for activations and weights and provide the empirical critical magnitudes reported below.

• For activation perturbations, when $|\epsilon| < 10^{-3}$, the first-order term explains over 90% of the observed loss shift, while the second- and higher-order terms contribute negligibly. When $10^{-3} \leq |\epsilon| < 8 \times 10^{-2}$, the second-order term's contribution becomes significant, warranting its inclusion if high approximation fidelity is desired. However, when $8 \times 10^{-2}$, higher-order terms become non-negligible, and the approximation loss of first- and second-order terms degrades rapidly.

• For weight perturbations, we observe a higher tolerance to noise. When $|\epsilon| < 8 \times 10^{-3}$, the first-order gradient term remains dominant, even though a well-trained model ideally has vanishing weight gradients. In practice, small but non-zero gradients persist and must be accounted for. When $8 \times 10^{-3} \leq |\epsilon| < 2 \times 10^{-1}$, second-order effects start to manifest, though still moderate. Only when $|\epsilon| < 2 \times 10^{-1}$ do higher-order terms begin to meaningfully affect the loss, primarily due to compounding curvature effects.

Despite the theoretical advantages of including second-order terms (e.g., curvature-aware approximations), we choose to truncate the expansion at the first order in our method. The decision is based on 3 key observations: (1) The marginal gain from second-order terms is often negligible—as confirmed by our experiments, where the second-order error contributed less than $10^{-5}$ to the loss; (2) In decomposition settings, low-rank approximations distort the original covariance structure, leading to unreliable Hessian estimates; (3) Second-order terms introduce substantial computational and memory overhead. LoD can give an operational first-order dominating radius $\varphi_{\max}$, which serves as an empirical threshold for determining whether the first-order approximation is valid.

> **Answer**: Within the first-order range, the Eq. 3 is updated to $\frac{1}{m}\sum\limits_{(x_j,y_j)\in\mathbb{D}}\sum\limits_{i=1}^{n}\frac{\partial\ell}{\partial h_{i+1}}\cdot\epsilon_i +$
> $\frac{\partial\ell}{\partial w_i}\cdot\delta_i$. Choosing $\epsilon$ and $\delta$ in the opposite direction of the corresponding gradients leads to a lower loss than the full-precision model. Concretely, for each component i, choosing $\epsilon_i = -\eta\,\mathrm{sign}(\frac{\partial\ell}{\partial h_{i+1}}), \delta_i = -\eta\,\mathrm{sign}(\frac{\partial\ell}{\partial w_i})$, with a suitably small compression step size $\eta$, ensures every inner product is negative. With sufficiently small perturbations, such gradient-opposing choices allow mixed precision compression to achieve stable gains over the original model.

## 2.3 THEORETICAL CHARACTERIZATION OF LOSSLESS COMPRESSION BOUNDARIES

To characterize the conditions for lossless compression, we model the noise as a perturbation vector $e$ applied to activations. Let $\mathbf{p} = \nabla_{h_{t+1}}\ell = [p_1, \ldots, p_k]^\top$ and $\mathbf{e} = [e_1, \ldots, e_k]^\top$, assuming $p_i$, $e_i$ i.i.d. entries and independence respectively, abbr. $p$, $e$. The induced loss change is approximated by $\Delta\ell \approx \mathbf{p}^\top\mathbf{e} = \sum_{i=1}^{k} p_i e_i$. Its expectation and variance satisfy

$$\mathbf{E}[\mathbf{p}^\top\mathbf{e}] = \sum_{i=1}^{k}\mathbf{E}[p_i]\mathbf{E}[e_i] = k\,\mathbf{E}[p]\,\mathbf{E}[e]. \tag{7}$$

The variance in general is

$$\mathrm{Var}(\mathbf{p}^\top\mathbf{e}) = \sum_{i=1}^{k}\mathrm{Var}(p_i e_i) = k\Big(\mathrm{Var}(p)\,\mathrm{Var}(e) + \mathrm{Var}(p)\,\mathbf{E}[e]^2 + \mathrm{Var}(e)\,\mathbf{E}[p]^2\Big), \tag{8}$$

To ensure an expected reduction in loss, we require $E[\mathbf{p}^\top \mathbf{e}] < 0$. In practice, this is achieved by selecting rounding directions or rank-dependent noise so that $\mathbf{E}[p]$ opposes $\mathbf{E}[e]$. Applying Chebyshev's inequality, we obtain a high-probability bound on failure to reduce loss:

$$P(\mathbf{p}^\top \mathbf{e} \geq 0) \leq \frac{\mathrm{Var}(\mathbf{p}^\top \mathbf{e})}{(\mathbf{E}[\mathbf{p}^\top \mathbf{e}])^2} = \frac{\mathrm{Var}(p)\mathrm{Var}(e) + \mathrm{Var}(p)\mathbf{E}[e]^2 + \mathrm{Var}(e)\mathbf{E}[p]^2}{k\,\mathbf{E}[p]^2\mathbf{E}[e]^2}. \tag{9}$$

This bound defines a lossless-compression regime: if the denominator dominates the numerator, the failure probability becomes negligible. Otherwise, the perturbation magnitude must be reduced (e.g., via higher bit-width or rank). For example, in INT8 quantization, each activation is mapped to one of 256 discrete levels. The per-element perturbation $e_i$ has extremely small variance $\mathrm{Var}(e) \approx \frac{(0.5/127)^2}{3}$. Typical activation gradients satisfy $\mathbf{E}[p] < 10^{-1}$. Taking channel counts $> 10^4$, the failure probability falls below $10^{-3}$.

## 3 LoD Quantization and Decomposition

**Decomposition.** Guided by LoD, we address decomposition as a rank-deficiency problem: the rank of each weight matrix is selected to minimize the loss shift within the differential neighborhood. For efficiency, we employ a low-rank factorization with an inequality constraint:

$$\min_{\delta^k} \bar{f}(w) - f(w) \approx \frac{1}{m} \sum_{i=1}^{n} \sum_{(x_j, y_j) \in \mathbb{D}} \left( g_i |\delta_i^k|_2 \cos\theta_i \right) + \lambda \sum_{i=1}^{n} \mathcal{N}(\delta_i^k) \tag{10}$$

We aim to minimize the additional loss induced by a low-rank perturbation $\delta^k = W - LR^\top$ in each layer. Specifically, the optimization seeks $\delta^k$ that minimizes the weighted projection onto the gradient, $\sum_i g_i |\delta_i^k|_2 \cos\theta_i$, where $g_i = ||\partial\ell/\partial w_i||_2$ and $\theta_i$ is the angle between $\delta_i^k$ and the gradient $\partial\ell/\partial w_i$, thereby encouraging perturbations along the gradient-negative direction to reduce loss. The perturbation is constrained within a parameter-wise neighborhood $|\delta_{i,j}^k| \leq \tau_{i,j}$, given in Fig. 2, and restricted to a low-rank subspace $\mathcal{S}_k = \mathrm{span}\{u_1, ..., u_k\}$ with $0 < k < rank_{max} = \frac{NM}{N+M}$ ($N$ and $M$ are the dimensions of the matrix), while an additional penalty term $\lambda \sum_i \mathcal{N}(\delta_i^k)$ controls the amplitude of each perturbation. This formulation jointly captures the directionality, magnitude, and rank properties of the perturbation to efficiently minimize loss.

---

**Algorithm 1** Layer-wise Optimal Rank Selection

---

**Input:** Neural network $M$ with $n$ layers, maximum rank $rank_{\mathrm{max}}$, tolerance $\tau$
**Output:** Optimal ranks $\{k_a\}$
1: **for** each layer $Layer_a$ in $M$ **do**
2:     Initialize candidate list $A \leftarrow \emptyset$
3:     **for** $c = 1, \ldots, rank_{\mathrm{max}}$ **do**
4:         Compute rank-$c$ subspace $\mathcal{S}_c = \mathrm{span}\{u_1, \ldots, u_c\}$
5:         Generate $\delta^c \in \mathcal{S}_c$ with $|\delta_{i,j}^c| \leq \tau$
6:         Align along gradient-negative direction: $\delta_i^c \leftarrow -\mathrm{sign}(\delta_i^c \cdot G_i) \cdot \delta_i^c$
7:         Compute projected loss $L(\delta^c)$
8:         **if** $L(\delta^c) < \epsilon$ **then**
9:             Record $c$ in $A$ and early stop
10:         **end if**
11:     **end for**
12:     Select optimal rank: $k_a = \min A$
13: **end for**
14: **return** $\{k_a\}$

---

Algorithm 1 generates candidate perturbations in each rank-$c$ subspace (obtained via SVD), aligning them with the negative gradient, evaluating the projected loss, and selecting the minimal rank satisfying the loss threshold $\epsilon$. This ensures that the chosen ranks both respect the low-rank structure and reduce loss, in accordance with LoD principles. For more implementation details, see the appendix.

**Quantization.** We propose a novel mixed precision quantization method grounded in the Loss-driven (LoD) framework, which addresses two key challenges in post-training quantization: (1) how

to achieve lossless or near-lossless quantization in mixed precision settings, and (2) how to efficiently select the optimal bit-width for each layer, a known NP-hard problem. For the first challenge, LoD quantization is applied for first-order analysis, ensuring lossless quantization within the first-order bounds. The second challenge is reformulated as a group knapsack problem, which is solved efficiently using dynamic programming. In the LoD framework, the loss function is treated as the "value $P$", each layer $i$ is considered a "group" with one bit-width $j$ choice per group, and the model size is treated as the "knapsack capacity $W$". This transforms the original problem into a low-computation group knapsack problem, where the goal is to select the optimal bit-width for each layer to minimize loss while keeping the quantized model size within the specified capacity $C$.

$$\min \sum_{i=1}^{n} P[i][j] \quad s.t. \sum_{i=1}^{n} W[i][j] < C, j \in [1, k], j \in \mathbf{Z} \tag{11}$$

where $n$ is the number of model layers. The problem scale of the grouped knapsack is very small, usually less than $n * k$, and has a significant efficiency advantage. The overall process of our proposed method is shown in Algorithm 2. In the algorithm, $\epsilon$ and $\delta$ denote the quantization noise of

---

**Algorithm 2** Lossless Mixed Precision Search Grouped Knapsack Algorithm

---

1: **Input:** Neural network $M$ with $n$ layers, quantization levels $[q_1, q_2, ..., q_k]$, maximum error $error_{max}$, calibration dataset $D$
2: **Output:** Cost matrix $P$, weight matrix $W$ of size $n \times k$
3: Calibrate network $M$ with dataset $D$ to collect data distribution
4: **for** each $q_j$ in $[q_1, q_2, ..., q_k]$ **do**
5:     **for** each $Layer_i$ in $M$ **do**
6:         Calculate model size $W[i][j]$ at $q_j$
7:         Compute $||\epsilon_i||$ and $scale_{input}$ for $Layer_i$
8:         Calculate $slope = \frac{f_{input}(M; scale_{input}, i) - f(M)}{scale_{input}}$
9:         Compute $fluc = f_{weight}(M; scale_{weight}, i) - f(M)$
10:         **if** $||fluc|| < error_{max}$ **then**
11:             Update $P[i][j] = slope \times \frac{||\epsilon_i||}{\sqrt{size(e_i)}}$
12:         **else**
13:             Compute $||\delta_i||$ and $scale_{weight}$
14:             Update $P[i][j] = slope \times \frac{||\epsilon_i||}{\sqrt{size(\epsilon_i)}} + \frac{fluc}{scale_{weight}} \times \frac{||\delta_i||}{\sqrt{size(\delta_i)}}$
15:         **end if**
16:     **end for**
17: **end for**
18: **return** $P, W$

---

activations and weights, and $Scale$ is the quantization parameter. The slope represents the gradient magnitude, $f(M)$ the loss of model $M$, and $f_{weight/input}(M; , noise, i)$ the loss when injecting noise into layer $i$'s weights or inputs. Quantization directions may be positive or negative. We set the quantization level in Algorithm 2 to $k = 4$ (2/4/8/16 bit), yielding a total complexity of $O(nk \cdot feature)$. Based on the $P$ and $W$ matrices, the optimization in Eq. 11 can then be solved via standard dynamic programming.

## 4 EXPERIMENT

### 4.1 DATASETS AND DETAILS.

**Datasets.** The ImageNet-1K dataset Krizhevsky et al. (2017) consists of 1.28 million training and 50K validation images. ImageNet-1K is usually used as the benchmark for model compression. The Stanford Question Answering Dataset (SQuAD) Rajpurkar et al. (2016) is a collection of question-answer pairs derived from Wikipedia articles. In SQuAD, the correct answers to questions can be any sequence of tokens in the given text. MNLI Williams et al. (2017) is a dataset for natural language reasoning tasks. Its corpus is a collection of textual implication annotations of sentences through crowdsourcing. The task is to predict whether the premise sentence and the hypothesis sentence are logically compatible (entailment, contradiction, neutral). MMLU Hendrycks et al. (2020) evaluates

large language models on 57 subjects across STEM, humanities, and social sciences, requiring broad knowledge and reasoning ability.

**Details.** The LoD scheme does not involve fine-tuning or retraining. We utilize the VGG Simonyan & Zisserman (2014), MobileNet Howard (2017), ResNet He et al. (2016) series (including ResNet-18, 34, and 50) to determine the error bounds depicted in Fig. 2. In the implementation, error bounds can be flexibly computed using Eq. 4 across various models on multiple datasets. Experiments show that, although the error bounds vary, the majority of models fall within this defined range. The parameters $error_{max}$ and $\gamma$ are set to approximately $10^{-4}$ in the algorithm. Quantization parameters are calculated using the ACIQ method. The validation set of ImageNet is used as the calibration set, where we check gradients without updating the weights. To ensure fairness, all experiments are conducted under identical optimization settings and executed on two NVIDIA A800 GPUs. The models are implemented based on pre-trained full-precision configurations in PyTorch.

### 4.2 ABLATION

**Compressed Noise Bounds.** Theoretical compression error bounds depend critically on model sensitivity to perturbations. High sensitivity restricts the compressible range, as large perturbations violate first-order approximation assumptions, impeding stable lossless compression.

To evaluate this, we calculate the upper bound of the first-order approximation error $N_{\epsilon^k}$ under different activation perturbations $\epsilon$ using Eq. 4. Fig. 3a reports the computed values across several representative models on ImageNet. These values reflect the deviation between actual loss and its first-order predicted counterpart. Smaller values indicate that first-order approximation is more accurate. Experiments demonstrate that for small noise (e.g., $\epsilon \leq 10^{-3}$), LoD's first-order estimate closely matches observed loss, indicating negligible second-order effects.

For instance, to ensure the loss change is smaller than $6 * 10^{-5}$ (the smallest positive number representable in FP16), it suffices to keep $\epsilon < \sqrt{0.00006}$. In this case, higher-order terms can be safely ignored and first-order estimates dominate. This suggests that the theoretical bounds not only reflect model robustness but also serve as a practical criterion for determining whether LoD-based quantization is appropriate for a given model.

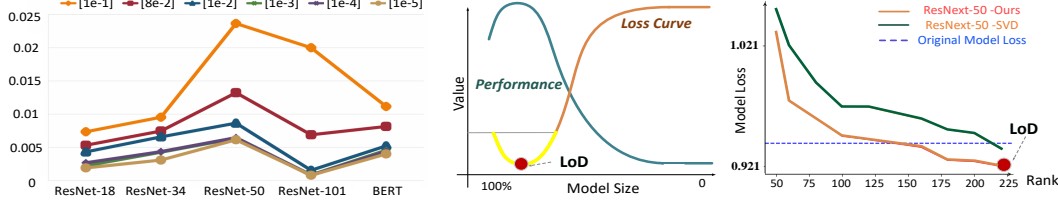

Figure 3: Left(a) Computed neighborhood $N_{\epsilon^k}$ under different activation noises $\epsilon$; color denotes noise level, y-axis shows neighborhood magnitude. Middle(b) and Right(c) LoD performance and loss curves for quantization and decomposition.

**Why 2-bit Quantization Is Avoided.** Although gradients in well-trained models are near zero, 2-bit quantization often introduces noise around $10^{-1}$—exceeding the first-order neighborhood and causing unstable neighborhood change in Fig. 3a. In contrast, 4-bit and 8-bit quantization induce much smaller noise ($< 5 \times 10^{-3}$), remaining within controllable bounds. Therefore, LoD primarily uses 4-bit and 8-bit for stability, selectively applying 2-bit only to layers with sufficient tolerance.

### 4.3 EVALUATION OF LoD

To rigorously assess the effectiveness of LoD, we perform decomposition and quantization experiments alongside standard benchmarks.

**Gains Brought by Decomposition.** As shown in Table 1, we apply LoD to decompose various models Liu et al. (2021b; 2022) on ImageNet, achieving consistent loss reduction across both convolutional and transformer-based architectures. Unlike quantization, decomposition changes the

Table 1: LoD-Decomposition consistently improves performance across both CV and NLP tasks. Acc. and Entropy represent accuracy and cross entropy (%). Compress indicates compression rate.

| Model / Task | Top-1 / Acc ↑ | Top-5 / EM / Acc ↑ | Entropy ↓ ±std | Compress. ↓ |
|---|---|---|---|---|
| Swin_S (ImageNet) | 81.08 → **81.13** | 95.61 → **95.61** | 81.19 → **81.00** ±0.004 | ↓43% |
| Swin_T (ImageNet) | **82.78** → **82.78** | 96.29 → **96.33** | 73.97 → **71.34** ±0.073 | ↓29% |
| VGG16 (ImageNet) | 69.20 → **69.43** | 88.90 → **88.94** | 114.54 → **114.22** ±0.011 | ↓67% |
| VGG19_BN (ImageNet) | 74.21 → **74.22** | 91.84 → **91.89** | 104.26 → **102.14** ±0.041 | ↓43% |
| ResNet-50 (ImageNet) | **76.13** → 76.10 | 92.86 → **92.90** | 96.18 → **95.05** ±0.016 | ↓56% |
| ConvNeXt_L (ImageNet) | **84.12** → **84.12** | 96.87 → **96.88** | 77.09 → **76.91** ±0.008 | ↓33% |
| BERT_base (SQuAD) | **85.74** → 85.67 | **80.49** → 80.42 | 44.61 → **44.60** ±0.000 | ↓45% |
| BERT_base (MNLI Val/Test) | 82.77 → **82.78** | 83.91 → **83.92** | 2.89 → **2.89** ±0.004 | ↓45% |
| TinyLlama (MMLU) | 26.93 → **27.01** | - | - | ↓22% |

Table 2: Comparison of Full-Prec (Full-Precision) and LoD-Quantized Models Across Tasks. LoD uses 2/4/8 bit mixed precision quantization. Acc and Entropy represent accuracy and cross entropy (%). Compress indicates compression rate, std represents standard deviation.

| Task | Model | Metric ↑ | Full Prec. | Ours | Entropy ↓ ±std | Compress. ↓ |
|---|---|---|---|---|---|---|
| MNIST | CNN | Top-1 | 97.51 | **97.66** | **7.92 → 7.86** ±0.019 | ↓73% |
| CIFAR | VGG13 | Acc. | 73.69 | **74.09** | **127.26 → 125.03** ±0.062 | ↓74% |
| | MobileNet | Acc. | 66.21 | **66.59** | **156.53 → 156.31** ±0.000 | ↓69% |
| | ResNet-14 | Acc. | 86.68 | **87.23** | **36.34 → 35.76** ±0.016 | ↓56% |
| | MobileNet_V2 | Acc. | 62.44 | **62.88** | **163.58 → 162.45** ±0.023 | ↓71% |
| ImageNet | VGG16_BN | Top-1/Top-5 | 73.34 / 91.51 | **73.71 / 91.52** | **106.62 → 105.43** ±0.009 | ↓66% |
| | MobileNet_V1 | Top-1/Top-5 | 70.28 / 89.43 | **70.84 / 89.68** | **114.79 → 114.66** ±0.014 | ↓68% |
| | MobileNet_V2 | Top-1/Top-5 | 71.89 / 90.29 | **71.89 / 90.30** | **114.80 → 114.78** ±0.003 | ↓71% |
| | ResNet-50 | Top-1/Top-5 | 75.06 / 92.42 | **75.09 / 92.44** | **100.19 → 98.54** ±0.082 | ↓66% |
| SQuAD | BERT | EM / F1 | 80.49 / 88.15 | **80.51 / 88.15** | **44.61 → 44.61** ±0.002 | ↓45% |

structure of weight matrices, making compression more sensitive and challenging. Despite this, LoD steadily lowers the loss while preserving Top-1/Top-5 accuracy, demonstrating its robustness.

Fig. 3c illustrates the performance and loss curves for LoD when compressing the ResNext-50 model. During decomposition, LoD identifies the lowest rank suitable for lossless compression. Compared to SVD methods, LoD more reliably identifies low-rank matrices that preserve accuracy, achieving effective model compression.

**Gains Brought by Quantization.** Table 2 shows that LoD quantization achieves lossless or improved performance across various tasks in both computer vision (CV) and natural language processing (NLP). Notably, in CV tasks like ImageNet and CIFAR-100, LoD successfully reduces model size with mixed precision quantization (e.g., 8/4/2-bit) while maintaining or enhancing accuracy. For instance, even 2-bit quantization is feasible for certain layers in models like VGG and MobileNet, thanks to their low sensitivity to quantization noise.

In NLP tasks, such as BERT on SQuAD, LoD applies more conservative 8-bit quantization, yet still achieves lossless compression with up to 45% storage reduction. This demonstrates LoD's robustness in maintaining stability in more sensitive tasks.

The primary goal is not just minimizing bit-widths, but ensuring theoretical stability under quantization. Our differential analysis shows that when quantization noise is aligned with the negative gradient, it can even reduce loss further, underscoring LoD's effective use of noise directionality.

**Comparisons.** Fig. 1 compares LoD with quantization and decomposition methods on ImageNet, showing existing approaches often achieve limited lossless rates (e.g., 2/6) with unstable gains Louizos et al. (2019); Chenna (2023); Zhang et al. (2025; 2023); Hu et al. (2021). In contrast, the proposed LoD method achieves a full lossless rate of 6/6 across all models, demonstrating superior stability while maintaining accuracy. For detailed comparison data, please see the Appendix.

Table 3 evaluates LoD against existing compression methods Wang et al. (2019); Liu et al. (2021a); Hu et al. (2023); Frantar et al. (2023); Zhang et al. (2024); Lin et al. (2024); Zhang et al. (2025)

Table 3: Compression results. Quant indicates LoD quantization. Convolution, Transformer and LLM use quantization. Drop indicates performance reduction. Compressed model outperforming origin yields negative values.

| | Model | Method | Orgin | Quant ↑ | Drop ↓ | Size |
|---|---|---|---|---|---|---|
| **Convolution (ImageNet)** | MobileNet-V2 | Multipoint | 71.78 / 90.19 | 70.70 / 89.70 | 1.08 / 0.49 | 2.09 MB |
| | | Hu et al. | 72.91 / 90.82 | 72.67 / 90.64 | 0.24 / 0.18 | 2.09 MB |
| | | HAQ | 71.87 / 90.32 | 71.85 / 90.24 | 0.02 / 0.08 | 2.09 MB |
| | | CET | 71.89 / 90.29 | 71.88 / 90.10 | 0.01 / 0.19 | 2.10 MB |
| | | **Ours** | **71.89 / 90.29** | **71.89 / 90.30** | **0.00 / -0.01** | **2.09 MB** |
| **Transformer (SQuAD1.1)** | BERT | FPxInt | 80.49 / 88.15 | 80.51 / 88.03 | -0.02 / 0.12 | 67.73 MB |
| | | **Ours** | **80.49 / 88.15** | **80.51 / 88.15** | **-0.02 / 0.00** | **63.20 MB** |
| **LLM (MMLU)** | TinyLlama | GPTQ | 26.93 | 26.01 | 0.82 | 1.80 GB |
| | | AWQ | 26.93 | 26.98 | -0.05 | 1.80 GB |
| | | **Ours** | **26.93** | **27.01** | **-0.08** | **1.80 GB** |

across convolutional (MobileNet-V2, ImageNet), transformer (BERT, SQuAD1.1), and large language model (TinyLlama, MMLU) architectures, using Top 1/5, EM/F1, and average score. Unlike other methods, which often suffer from performance degradation or inconsistent gains, LoD achieves lossless or improved accuracy across all models, significantly reducing model sizes while maintaining stability.

Table 4 compares LoD's first-order approximation with second-order and full Hessian inversion on ResNet-50, showing LoD's 0.7 s/layer and $\Delta$Loss of $O(10^{-3})$ versus 450 s ($O(10^{-6})$) and 30,000 s ($O(10^{-8})$) for higher-order methods. This justifies LoD's first-order truncation, as its $\Delta$Loss ensures stable, lossless compression (6/6 rate, Table 3). Higher-order terms are impractical due to complexity and instability. Using differential neighborhood analysis, LoD controls nonlinear fluctuations. A more detailed second-order discussion is provided in the Appendix. Moreover, LoD's training-free design enables quantization in under 5 minutes and decomposition in 15 minutes, enhancing deployment efficiency across diverse architectures.

Table 4: Comparison of approximation methods on ResNet-50. Time measures computational cost per layer, $\Delta$**Entropy** quantifies cross entropy change approximation. LoD's first-order approach balances efficiency and stability, supporting its use in mixed precision compression.

| Method | Time | $\Delta$**Entropy** | Notes |
|---|---|---|---|
| LoD | $\sim 0.7$ s | $O(10^{-3})$ | Gradient |
| Second-Order | $\sim 450$ s | $O(10^{-6})$ | $\sim$300 Hessian–vector products |
| Full Hessian | $\sim 3 \times 10^4$ s | $O(10^{-8})$ | $O(n^3)$ Inversion |
| Higher-Order | Intractable | $< O(10^{-8})$ | Impractical due to computational complexity |
| HAQ | 384 h | $O(10^{-3})$ | Quantization-aware training |

**Limitations.** LoD leverages differential neighborhood analysis (Eq. 4) to explain performance gains in mixed-precision compression. However, in extreme compression scenarios, such as 2-bit quantization or extremely low-rank approximations, large compression errors ($\delta/\epsilon$) render higher-order derivative terms non-negligible. Furthermore, existing extreme compression techniques have yet to achieve lossless performance. These factors introduce uncertainty, preventing stable performance improvements and exceeding LoD's first-order analysis scope. In the appendix, we provide detailed limitations of LoD compression, as well as experiments and analysis of the second-order term.

## 5 CONCLUSION

This work challenges the view that compression degrades performance, often deemed fortuitous. We provide a systematic theoretical analysis of mixedprecision compression gains, using differential neighborhoods to bound first- and higher-order effects for stable, lossless outcomes. Our LoD framework, applied to quantization and decomposition, consistently achieves interpretable, modelagnostic improvements across diverse tasks and architectures, as validated by extensive results.

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
