# APPENDIX

## A  APPENDIX

### A.1  RELATED WORKS

**Tensor Decomposition.**  Conventional post-training tensor and matrix decomposition techniques (e.g., SVD, CP, Tucker) compress models by factorizing weight tensors into low-rank components. Without subsequent fine-tuning, such decompositions frequently increase loss substantially and require retraining to recover accuracy. Representative works include methods that combine low-rank factorization with feature-map reconstruction or training-time regularization to mitigate degradation Yu et al. (2017); Xu et al. (2019); Yang et al. (2020); Zhang et al. (2023). Recent approaches (e.g., DAC, MAESTRO) have reported near-lossless outcomes in some settings, though often relying on additional retraining or specialized ordering of factors Li et al. (2019); Horváth et al. (2024). Our LoD framework differs from these in that it (i) does not depend on retraining to ensure loss-preservation, and (ii) provides a differential (loss-driven) criterion to select per-layer ranks within an empirically validated local neighborhood where first-order analysis is predictive.

**Mixed Precision Quantization.**  Post-training quantization maps model weights and activations to lower bit-width representations without full retraining Banner et al. (2018); Li et al. (2021); Dong et al. (2020); Wang et al. (2019); Lou et al. (2019); Zhang et al. (2024). Mixed-precision strategies assign different bit-widths to different layers according to sensitivity analyses; several methods automate this search via reinforcement learning or search heuristics (e.g., HAQ, AutoQ) Wang et al. (2019). More recently, works such as DFQ Xu et al. (2020), RQ Louizos et al. (2019), and geometric allocation schemes like CET Zhang et al. (2025) have reported cases of near-lossless or even slight accuracy improvements under particular settings. LoD complements this line by formalizing when such gains can be expected from the perspective of local loss perturbation theory.

Our LoD framework differs in three key aspects. First, LoD eliminates the need for post-training and retraining: While most decomposition and mixed-precision quantization methods require fine-tuning or iterative adjustments to recover accuracy, LoD utilizes only a small calibration set for compression decisions, avoiding any additional retraining costs. Second, LoD provides a clear theoretical criterion and explanation based on differential perturbation analysis: By expanding the loss differential, LoD introduces a testable neighborhood radius $\phi_{\max}$ that describes when higher-order analysis is used and when compression noise is inversely proportional to gradient information, resulting in loss preservation or reduction. Existing methods rely primarily on heuristics (e.g., sensitivity metrics, reinforcement learning search) but lack such formal guarantees and theoretical explanations for the performance gains after compression. Third, LoD unifies tensor decomposition and mixed-precision quantization under the same loss-driven framework: rank selection and bitwidth allocation are both transformed into optimization problems on a local loss bound, while quantization is further instantiated as a grouped knapsack formulation. This unifying perspective not only clarifies when "lossless" compression can be achieved, but also provides viable algorithms that connect previously unrelated research threads in decomposition and quantization.

### A.2  LOD

#### A.2.1  ALGORITHM DETAILS

The Layer-wise Optimal Rank Selection algorithm identifies the minimum feasible rank for each layer that satisfies a fixed tolerance of $\tau = 10^{-2}$. The procedure starts by incrementally expanding the candidate subspace from rank-1 up to $rank_{\max}$. At each candidate rank $c$, a perturbation $\delta^c$ is constructed within the corresponding subspace, with element-wise magnitude bounded by $|\delta^c_{i,j}| \leq \tau$

to ensure consistency of perturbation scale across ranks. Each perturbation is then aligned with the negative gradient direction, ensuring that the first-order effect on the loss is non-increasing. The search terminates as soon as a candidate rank meets the tolerance condition, and this rank is recorded as the optimal choice for the given layer.

The Lossless Mixed Precision Search Grouped Knapsack Algorithm formulates bit-width assignment as a grouped knapsack problem, where each layer is assigned one candidate precision from $2, 4, 8, 16$ with an associated storage cost $W[i][j]$ and performance impact $P[i][j]$. The procedure first calibrates the network with a small dataset to estimate activation and weight distributions, then for each layer and quantization level computes model size, activation noise $\epsilon_i$, weight noise $\delta_i$, and gradient slope, updating $P[i][j]$ with normalized noise contributions. The DP formulation then selects one candidate per layer to minimize the global loss increase under storage constraints. Following existing methods, the calibration set selects an existing dataset, such as the validation set of ImageNet. The formula $slope = \frac{f_{input}(M; scale_{input}, i) - f(M)}{scale_{input}}$ effectively approximates the sensitivity of a model's loss to input perturbations at layer $i$, resembling the secant line slope between noisy points $f(M)$ and $f_{input}(M; scale_{input}, i)$. It enables analysis without requiring analytical gradients, making it ideal for black-box models or post-training compression studies.

### A.2.2 WHY THE DENOMINATOR CAN DOMINATE IN EQ.9

The intuition is statistical: for $X = p^\top e = \sum_{i=1}^{k} p_i e_i$ (with independent channel entries) the mean and variance scale differently with the channel count $k$. Under the i.i.d. assumptions used above,

$$\mu = \mathbf{E}[X] = k\,\mathbf{E}[p]\,\mathbf{E}[e], \qquad \mathrm{Var}(X) = k\,V,$$

where $V = \mathrm{Var}(p)\mathrm{Var}(e) + \mathrm{Var}(p)\mathbf{E}[e]^2 + \mathrm{Var}(e)\mathbf{E}[p]^2$. Thus the Chebyshev upper bound can be written as

$$P(X \geq 0) \leq \frac{\mathrm{Var}(X)}{\mu^2} = \frac{V}{k\,\mathbf{E}[p]^2\mathbf{E}[e]^2}.$$

Two simple scaling observations follow. First, the numerator $V$ does not grow with $k$ (it is per-channel), while the denominator scales as $k$; hence increasing channel count reduces the ratio roughly as $1/k$. Second, the signal-to-noise ratio

$$\mathrm{SNR} = \frac{|\mu|}{\sqrt{\mathrm{Var}(X)}} = \sqrt{k}\,\frac{|\mathbf{E}[p]\mathbf{E}[e]|}{\sqrt{V}}$$

grows like $\sqrt{k}$, so tail probabilities decay rapidly with $k$ (under a normal/CLT approximation, the tail probability falls approximately like $\exp(-\frac{1}{2}\mathrm{SNR}^2)$, i.e. exponentially in $k$).

As an illustrative plug-in (using numbers of the same order as discussed above), take $\mathbf{E}[p] = 0.1$, $\mathbf{E}[e] = 3.9 \times 10^{-3}$, $\mathrm{Var}(p) = 10^{-2}$, and $\mathrm{Var}(e) \approx (0.5/127)^2/3 \approx 5.17 \times 10^{-6}$ with $k = 10^4$. Then $V \approx 2.55 \times 10^{-7}$, $\mu = k\mathbf{E}[p]\mathbf{E}[e] \approx 3.9$, and $\mathrm{Var}(X) = kV \approx 2.55 \times 10^{-3}$. The Chebyshev bound gives $P(X \geq 0) \lesssim \frac{2.55 \times 10^{-3}}{3.9^2} \approx 1.7 \times 10^{-4}$, (i.e. $< 10^{-3}$). Under the CLT the standardized mean is $\mu/\sqrt{\mathrm{Var}(X)} \approx 77$, producing a numerically negligible tail probability. This demonstrates that, when (i) channel count $k$ is large and (ii) the per-channel mean product $\mathbf{E}[p]\mathbf{E}[e]$ is non-negligible, the accumulated deterministic signal (numerator of $\mu$) overwhelms random fluctuation (variance), making the Chebyshev/CLT bounds very small.

This conclusion depends on key assumptions: independence (or weak correlation) across channels, and a nonzero mean alignment $\mathbf{E}[p]\mathbf{E}[e] \neq 0$. If $\mathbf{E}[e] \approx 0$ (unbiased quantization) or strong channel correlations exist, the denominator may not dominate and the bound becomes uninformative. In practice we therefore (a) measure $\mathbf{E}[p], \mathbf{E}[e], \mathrm{Var}(p), \mathrm{Var}(e)$ on a calibration set, (b) compute the SNR and the Chebyshev/CLT estimates, and (c) when in doubt perform a small empirical forward-check on calibration data. These steps give an operational test for whether the analytic regime (denominator domination) applies.

Our work is motivated by a widely observed phenomenon: models often retain or even improve performance after compression, without fine-tuning. LoD provides a theoretical explanation for this by modeling the expected directional reduction in loss under compression noise perturbations. However, this explanation is not universally valid. The underlying assumptions of LoD—such as

Table 1: Layer-wise loss after quantization at different bit-widths. The baseline loss is 1.2726. 8-bit and 4-bit quantization have negligible impact, whereas 2-bit causes mild degradation (notably for layer #4), and 1-bit leads to a large loss increase.

| Layer | 8-bit | 4-bit | 2-bit | 1-bit |
|-------|-------|-------|-------|-------|
| #8 | 1.2689 | 1.2603 | 1.2599 | 2.1599 |
| #4 | 1.2711 | 1.2709 | 1.9961 | 2.6634 |

first-order approximability, gradient stability—do not always hold. Below, we outline scenarios where LoD's explanatory power may break down:

1) LoD assumes that the impact of compression on the loss can be captured by a first-order differential approximation. When the compression is too aggressive, the perturbation may fall outside the valid neighborhood, where the first-order terms + higher-order terms dominate, invalidating the theoretical prediction. At this time, LoD operates in a best-effort manner: if a layer can tolerate aggressive compression (i.e., ultra-low bits) without violating the neighborhood condition, the algorithm will compress it accordingly. For example, consider the VGG13 network. The table below presents the loss values when applying LoD to the 4th and 8th layers with different quantization bit-widths.

As shown in Table 1, 1-bit quantization introduces a loss that exceeds the allowable neighborhood for both layers, making it unacceptable. LoD therefore attempts 2-bit quantization. At this setting, the 8th layer remains within the neighborhood and even achieves a performance gain, while the 4th layer still experiences excessive loss. Consequently, LoD chooses 4-bit quantization for the 4th layer. This example illustrates that layers with wider error tolerance neighborhoods benefit more from LoD compression.

2) The effectiveness of LoD hinges on the expectation that the inner product between the gradient and the perturbation is negative. If the gradient is hard to estimate—such as when using unlabeled data—this directional descent assumption no longer holds, limiting LoD's predictive ability.

In addition, the mechanism revealed by LoD may also be exploited in other methods in a non-explicit form, although these methods do not explicitly model the directional principle. The contribution of LoD is that it first reveals the necessary conditions behind this phenomenon from a theoretical perspective and provides a unified analytical framework that can be used to guide and explore the performance of models after compression.

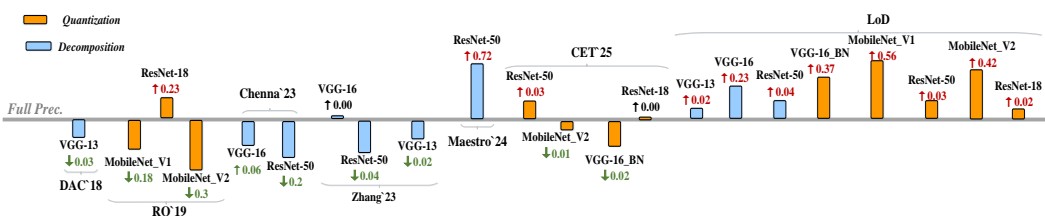

Figure 1: Performance and loss curves of LoD in quantization and decomposition. LoD achieves better performance with smaller models.

### A.2.3 IMPACT OF SECOND-ORDER TERMS UNDER PERFORMANCE GAINS

While second-order information is traditionally regarded as offering higher theoretical fidelity in modeling loss perturbations, we find its practical value in explaining performance gains from compression to be minimal. In modern deep networks, the perturbations introduced by compression are typically small. As a result, the loss change is well-characterized by first-order approximations, with second-order terms contributing only marginal corrections.

Furthermore, the cost of computing second-order derivatives is often prohibitive. Even efficient approximations, such as those based on the Lanczos algorithm, require several orders of magnitude more time than first-order gradients—e.g., hundreds of seconds per layer—while refining the loss

estimate by only $\mathcal{O}(10^{-6})$. Full Hessian inversion is practically restricted to toy layers and is entirely infeasible for large-scale networks.

More critically, second-order terms used in practice are seldom exact. Common strategies such as diagonal or low-rank approximations inevitably introduce estimation errors, undermining their purported precision. This, coupled with the fact that most mainstream compression methods (e.g., low-rank decomposition or quantization) are not explicitly designed to leverage second-order information, further limits their utility.

As summarized in Table 4 in original paper, first-order methods strike a superior balance between cost and accuracy, accounting for the vast majority of loss variation while remaining computationally tractable. In contrast, second- and higher-order methods, though mathematically rigorous, are computationally intractable and numerically negligible in real-world scenarios. Therefore, LoD adopts first-order analysis as a principled and practical foundation to explain performance gains under compression, emphasizing that higher-order terms do not substantively influence the observed effect.

To demonstrate the specific improvements achieved by LoD, Figure 1 illustrates the performance and loss curves of the proposed LoD method under quantization and decomposition compression strategies. Overall, LoD consistently improves the post-compression performance across various models and datasets, demonstrating strong robustness and effectiveness. Firstly, in the quantization section (orange bars), although some models experience a slight drop in accuracy after compression (e.g., ResNet-18 decreased by 0.23), most models maintain or even improve their performance with LoD applied (such as MobileNet_V2, improved by 0.56). This indicates that LoD effectively minimizes performance degradation during quantization.

Secondly, in the decomposition section (blue bars), LoD not only compresses the models but also stably enhances their accuracy. For example, Maestro'24, MobileNet_V2, and VGG-16_BN show accuracy improvements of 0.37, 0.01, and 0.02, respectively, after decomposition. In contrast, traditional methods like ResNet-50 tend to have either performance drops or marginal gains, highlighting the advantage of LoD.

Moreover, LoD yields more pronounced improvements on smaller models (e.g., MobileNet and VGG-16 variants), indicating its particular suitability for lightweight architectures. LoD consistently boosts compressed model performance under varying compression techniques and architectures, especially by stabilizing the training loss and increasing accuracy while maintaining model compactness.

## B  THE USE OF LARGE LANGUAGE MODELS (LLMs)

This paper:

1. uses LLM to identify grammatical errors, inappropriate expressions, and linguistic incoherence in texts and helps improve writing.

2. uses LLM to locate key literature in a specific research field during the search for relevant work.

3. uses LLM to check for errors in notation and standardization in formulas.