# OpenReview forum: "LoD: Unlocking Performance Gains in Compression via Differential Analysis"
_ICLR.cc/2026/Conference — ICLR 2026 Conference Withdrawn Submission_

### Official Review · Reviewer_wysd · 2025-10-28

**Soundness:** 2
**Presentation:** 2
**Contribution:** 2
**Rating:** 2
**Confidence:** 3

**Summary:**

The paper introduces Loss-Driven differentiable framework that models the loss change under model compression using a Taylor series expansion. By analyzing the expansion terms, the authors derive bounds on the permissible compression noise that define the lossless regime. For low-rank parameter decomposition, the proposed algorithm uses the gradient of the loss w.r.t. weights on a calibration set and a perturbation is a low-rank subspace to chose the rank corresponding to the lowest loss value. Compression is then applied in a single step without any further fine-tuning. The approach is extended to a bitwidth allocation for quantization via a knapsack algorithm, and evaluated on vision (CNN) and language (Transformer) models.

**Strengths:**

1. The one-shot compression procedure efficiently uses gradient information from a small calibration batch, without the need for fine-tuning.
2. The framework unifies low-rank decomposition and quantization based methods.
3. Empirical results demonstrate performance retention (or improvements over the full precision baseline) across vision and language domains.
4. The paper reports the actual achieved compression rates, most substantial for the smaller models.

**Weaknesses:**

1. Limited practical value: compression rates produced by Algorithms 1 and 2 are only bounded via the maximum decomposition rank $rank_{max}$ and maximum bitwidth $q_k$, respectively. Neither algorithm has estimates on the resulting compression ratio or loss gains.
This behavior is undesirable in deployment, where total memory is typically fixed, and one aims for the best performance within that constraint. As noted by the authors, the algorithm tends to assign 8-bit (=higher) precision for large models (e.g., BERT), reducing its relevance for practical scenarios.
2. The experimental validation is limited to relatively small scale models. Reported compression ratios are most impressive for smaller models, and are modest for larger ones (≈ 45 % for BERT, 22 % for TinyLlama), suggesting that the method’s assumptions and results degrade with scale. More broad testing is needed to show the claimed generality of the approach.
3. The used theoretical assumptions are overly simplistic: the analysis treats weight and activation perturbations of different layers as independent, which overlooks the error propagation through the model. Also, ignoring second-order effects limits the scope of the analysis, especially at aggressive compression levels (e.g., extreme low-bit quantization), where curvature terms dominate.

**Questions:**

1. In Algorithm 1, do you average the projected loss over several perturbations?
2. How does the size of the calibration set affect the results?
3. Do the reported performance and compression trends hold for larger model families such as ViT or larger LLMs?

---

### Official Review · Reviewer_af1S · 2025-10-29

**Soundness:** 3
**Presentation:** 2
**Contribution:** 3
**Rating:** 6
**Confidence:** 3

**Summary:**

The authors propose to analyze the compression of neural networks as optimizing the impact of the model loss instead of minimizing the weight perturbation. They provide a theoretical framework to analyze the impact of small weight perturbation by using a first-order approximation. They then show how this analysis can be used to choose layer-wise compression hyperparameters either for low-rank decomposition, or quantization under a certain memory budget. This method and analysis holds for few-bits (4 or more) quantization, although it doesn’t extend to higher quantization rates.

**Strengths:**

- The authors introduce a theoretical analysis and derive an expression to express bounds over the degradation in loss, and constraints to preserve or improve model capabilities under compression constraints.
- The method is evaluated over multiple tasks and modalities, and show consistent results with preservation of model capabilities in almost every setting.

**Weaknesses:**

The main weakness is related to the comparison with existing methods on quantization. This quantization scheme is applied to mostly small models, which is not where it is mostly useful. Comparison on medium-sized LLM (LLaMA-3 8B / Mistral 7B, or similar), and including other existing methods such as VPTQ, would strengthen the results.

Minor comments
- “when $8x10^2$” (line 180): typo, missing $|\epsilon|$
- “Answer:” block (line 196): Adding a red ‘answer’ is confusing and unnecessary.
- Figure 2 is confusing, it is not clear what it represents
- Figure 1 is suitable as an illustration for introduction, not as a reference for experimental results (line 426).
- Overall, writing could be made more clear, especially given the amount of analysis in the paper.

**Questions:**

Could the author provide more comprehensive results on settings with challenging quantization and broader applications?

---

### Official Review · Reviewer_VSjf · 2025-10-29

**Soundness:** 3
**Presentation:** 2
**Contribution:** 3
**Rating:** 4
**Confidence:** 2

**Summary:**

This paper looks into the phenomenon that mixed-precision compression can occasionally improve model performance rather than merely trading accuracy for efficiency. The authors propose LoD, a theoretical framework based on total differential analysis to systematically explain and achieve lossless compression. The framework addresses three key questions: conditions enabling performance gains, modeling performance instability, and theoretical boundaries for lossless compression. LoD is instantiated on two compression methods: tensor decomposition and quantization. Experiments across CV and NLP tasks demonstrate consistent lossless or improved accuracy with significant compression rates.

**Strengths:**

- Problem Formulation: The paper tackles an important and underexplored question, why compression can sometimes improve performance. By framing compression as an optimization process that perturbs parameters along the opposite direction of the gradient, the work addresses a known but insufficiently explained phenomenon
- Generalizability: The LoD framework is model-agnostic. The authors successfully deploy it across decomposition and quantization which are two principal compression paradigms.
- Practicality: In particular, the grouped knapsack algorithm for quantization provides an efficient and theoretically grounded solution to the mixed-precision search problem.

**Weaknesses:**

- Insufficient Validation: The key value LoD is its applicability to today’s most important and computationally intensive models, yet the paper’s experiments avoid these models, seriously questioning the generalizability of its conclusions.
  - Inadequate LLM Coverage: Language Model experiments are limited to "toy models" like BERT-base (110M) and TinyLlama (1.1B). These models are not representative of the current LLM landscape, and there is a complete lack of validation on mainstream large models like LLaMA-7B, OPT. Prior works such as SmoothQuant[1] and OmniQuant[2] perform comprehensive evaluations on large-scale models.
  - Lack of Generative Task Evaluation: The paper only evaluates on NLU tasks (SQuAD, MNLI, MMLU) and completely omits generative tasks. It does not assess performance on language modeling benchmarks such as WikiText2, PTB, or C4. These tasks are typically measured by perplexity (PPL) and are far more sensitive to the cumulative errors introduced by compression. Evaluation on these tasks is considered a good representation of  PTQ assessment.
- Insufficient Ablation Studies: The paper lacks critical ablation studies that undermine a thorough validation of the method’s robustness. Addressing gaps below with more ablation studies will strengthen the validation of the method’s effectiveness and reliability.
  - Lack of analysis of how individual components such as decomposition, grouped knapsack optimization, or dynamic bit/rank selection contribute to overall performance and stability.
  - Does not present sensitivity experiments on the neighborhood boundary parameter `error_max` to show how its settings affect results.

[1] SmoothQuant: Accurate and Efficient Post-Training Quantization for Large Language Models.

[2] OmniQuant: Omnidirectionally Calibrated Quantization for Large Language Models.

**Questions:**

See the weaknesses section.

---

### Official Review · Reviewer_m4MB · 2025-11-01

**Soundness:** 2
**Presentation:** 3
**Contribution:** 2
**Rating:** 4
**Confidence:** 2

**Summary:**

The paper suggest that compressing a neural network may improve its accuracy, unlike the common assumution that it entails performance degradation.
The authors suggest LoD (Loss-driven Differential Analysis) that uses mixed-precision compression (through quantization and decomposition) which is claimed to, under well-defined conditions, improve model accuracy rather than harm it.

The paper suggests a differential analysis framework to explain this observation, and expanding the loss function through total differentials. This yields higher-order approximations that characterize how compression noise affects loss. The following two practical instantiations are then introduced.
LoD-Decomposition optimizes per-layer rank selection under low-rank constraints, and LoD-Quantization reformulates bit-width allocation as a grouped knapsack optimization problem to achieve lossless compression within a bounded perturbation region.

Empirical results on ImageNet, SQuAD, MNLI, and MMLU show that LoD consistently achieves lossless or improved accuracy across CNNs, Transformers, and LLMs, outperforming existing compression methods like HAQ, GPTQ, AWQ, CET, and Maestro.

**Strengths:**

Conceptual originality - LoD reinterprets model compression as a controlled differential process rather than an empirical trade-off, offering a theoretical explanation for observed performance gains under compression.

Mathematical rigor and interpretability - the paper connects connects compression perturbations with total differentials of the loss, providing a principled foundation.

The introduction of the perturbation neighborhood and stability bounds in Eq.6 bridges theory and measurable quantities, and the probabilistic bound in Eq. 9 formalizes conditions for lossless compression using Chebyshev’s inequality.

Efficient implementation - The framework translates directly into algorithms (Algorithm 1 for decomposition and Algorithm 2 for quantization) that are model-agnostic and training-free, allowing fast compression without retraining.

Empirical evaluation - Evaluations span multiple data modalities (CV, NLP, LLMs) and architectures (ResNet, Swin, MobileNet, BERT, TinyLlama).

**Weaknesses:**

Assumptions- The dismissal of higher order terms limits the validity of the algorithm. The authors acknowledge this but could better quantify where the approximation breaks down.

Simplified noise modeling - The independence assumption between activation and gradient perturbations may not hold in practice, especially in transformer layers with strong interdependencies.

Comparison to training-based methods - It would be nice to compare against retraining-based schemes such as QAT or low-rank adaptation methods such as LoRA.

Theory - while there are a lot of equations, the theory is not so deep.

**Questions:**

In Eq(3), can we add more terms to obtain better accuracy?

---

### Official Review · Reviewer_Ffbp · 2025-11-06

**Soundness:** 1
**Presentation:** 1
**Contribution:** 2
**Rating:** 0
**Confidence:** 3

**Summary:**

The authors study the effects of weight decomposition and mixed-precision post-training quantisation in neural networks. They first attempt to characterise the quantisation error mathematically. Then, based on their results, they propose a weight decomposition and a mixed-precision quantisation algorithm. The core idea behind both the authors' theoretical ideas and their proposed algorithms is to perform lossy model compression so that the errors introduced in the activations of one layer cancel out the errors introduced in the weights of the next layer. The authors perform experiments on several computer vision and language modeling tasks, demonstrating that they can reduce model size while retaining high accuracy.

**Strengths:**

The high-level idea of having errors at various points in the network architecture cancel each other out is nice. I found it quite nice that the authors could reduce the model sizes in their experiments without affecting accuracy too much.

**Weaknesses:**

Unfortunately, the paper feels like it has a few nice ideas buried under a lot of difficult-to-understand and erroneous theory.

Although the authors never define their loss-driven (LoD) theoretical framework, I believe they primarily refer to Eq. 3 and the implications it entails. Sadly, Eq. 3 does not make sense:
 - First, the left-hand side minimises over $\epsilon$ (over some undefined set $E$), while on the right-hand side $\epsilon$ is present.
 - Second, the $\epsilon_i$s and $\delta_i$s are not defined as multiplicative errors in Eq 2, so it is unclear how they are multiplying the partial derivatives in Eq 3.

I did my best to try to guess in good faith what the authors might have meant in the rest of the paper, though I must admit, I definitely struggled.

Please find below a list of concrete problems I have with the writing; some are more related to the writing, some more to the framing, some to the theoretical part:
- L39: "Conventional wisdom views compression as inherently lossy" -- I am not sure which conventional wisdom the authors are referring to; compression can also be lossless.
- "with academia often attributing" -- "academia" sounds strange in this context, perhaps the authors mean "researchers"?
- "with academia often attributing such gains to fortuitous alignments" -- I have worked in the compression community for several years and have never heard such an attribution. Do the authors have some references to back this up?
- "To address this, we propose LoD, a unified loss-driven differentiable framework, focusing on three key questions" -- this sentence is repeated twice.
- Eq 1: $model(x)$ doesn't depend on $x$ in the definition, I believe we should have $x$ instead of $h_{n + 1}$ in the definition. Furthermore, in the definition of the loss, $model$ takes two arguments, while in the definition on the right, it only takes one argument.
- Eq 2 is difficult to read, and it's not clear why $h_{n + 1}$ is being perturbed. Furthermore, what is $L$?
- "This perspective enables the analysis of compression effects using perturbation theory and differential." -- I don't understand the second part of the sentence
- Eq 4: $w$ is not a bound variable on the right-hand side; is $\sup_w$ missing?
- Figure 2 shows results as if they were universal. However, this is a specific empirical observation, but there is no description of the experiments.
- L196: "Answer:" -- what question is being answered here? The color code does not match the color code of the questions from Section 1
- Section 2.3: Why are the authors suddenly switching to a multiplicative noise model? Furthermore, the assumption that the quantisation noise is independent of the source is completely unjustified, let alone the assumption that the gradient entries are iid!
- L231: "Guided by LoD" -- The authors haven't defined what LoD is yet.
- Eq 10: the use of $\theta$ is conflicting with $\theta$ from Definition 1
- It's unclear how Algorith 1 solves Eq. 10
- "Quantization parameters are calculated using the ACIQ method" -- citation please.
- Figure 3: Several missing axis labels.
- Fig 3b is a drawing, not a plot of experimental results; is this correct?
- Fig 3c: curve color doesn't match legend text
- Fig 3a: There are 6 legends given, but it looks like there are only 5 curves.
- Table 2: "Full Prec.", "Ours" and "Compress." columns: what are the numbers?

**Questions:**

n/a

---

### Note · Authors · 2025-11-14

I have read and agree with the venue's withdrawal policy on behalf of myself and my co-authors.